# Goal Recognition Design for General Behavioral Agents using Machine Learning

**Robert Kasumba**                                                                              *rkasumba@wustl.edu*
*Washington University in Saint Louis*

**Guanghui Yu**                                                                                *guanghuiyu@wustl.edu*
*Washington University in Saint Louis*

**Chien-Ju Ho**                                                                                *chienju.ho@wustl.edu*
*Washington University in Saint Louis*

**Sarah Keren**                                                                                *sarahk@cs.technion.ac.il*
*Technion – Israel Institute of Technology*

**William Yeoh**                                                                               *wyeoh@wustl.edu*
*Washington University in Saint Louis*

**Reviewed on OpenReview:** *https://openreview.net/forum?id=GDuWBhvMid*

## Abstract

*Goal recognition design (GRD)* aims to make limited modifications to decision-making environments to make it easier to infer the goals of agents acting within those environments. Although various research efforts have been made in goal recognition design, existing approaches are computationally demanding and often assume that agents are (near-)optimal in their decision-making. To address these limitations, we leverage machine learning methods for goal recognition design that can both improve run-time efficiency and account for agents with general behavioral models. Following existing literature, we use *worst-case distinctiveness* (*wcd*) as a measure of the difficulty in inferring the goal of an agent in a decision-making environment. Our approach begins by training a machine learning model to predict the *wcd* for a given environment and the agent behavior model. We then propose a gradient-based optimization framework that accommodates various constraints to optimize decision-making environments for enhanced goal recognition. Through extensive simulations, we demonstrate that our approach outperforms existing methods in reducing *wcd* and enhances runtime efficiency. Moreover, our approach also adapts to settings in which existing approaches do not apply, such as those involving flexible budget constraints, more complex environments, and suboptimal agent behavior. Finally, we conducted human-subject experiments that demonstrate that our method creates environments that facilitate efficient goal recognition from human decision-makers.

## 1 Introduction

With the rapid advancement of artificial intelligence (AI), there has been a surge in interest in human-AI collaboration, aiming to synergize human and AI capabilities across various domains, such as gaming, e-commerce, healthcare, and workflow productivity. Designing AI agents to work alongside humans requires these agents to understand and infer human goals and intentions. While there has been abundant research in goal recognition (Ramírez & Geffner, 2010; Sukthankar et al., 2014), aiming to infer human goals by observing their actions, this work focuses on the goal recognition design problem (Keren et al., 2014), which seeks to identify how to modify decision-making environments to make it easier to infer human goals.

Our work is built on the *GRD* problem formulated by Keren et al. (2014). They proposed the worst-case distinctiveness (*wcd*) metric, defined as the maximum number of decisions an agent can make without revealing its goal, to measure the difficulty of inferring the agent's goal. They then aimed to modify the decision-making environment, by removing allowable actions from states, to optimize this measure. Since the introduction of this work, there have been several follow-up works to deal with different settings, such as stochastic settings (Wayllace et al., 2016; 2017) and partial observability (Keren et al., 2016). More discussion can be found in the survey by Keren et al. (2021).

While there has been significant progress in *GRD*, there are two main limitations in the literature. First, existing approaches often assume that decision-making agents are optimal or near-optimal decision makers (Keren et al., 2015; Wayllace & Yeoh, 2022). This assumption is unrealistic with human decision makers, who are known to often systematically deviate from optimal decision-making due to cognitive and informational constraints (Kahneman, 2003; O'Donoghue & Rabin, 1999). Second, existing approaches require evaluating the difficulty of goal recognition, i.e., worst-case distinctiveness (*wcd*), for a large number of modifications to the environment. This computational requirement limits the scalability of these approaches, especially for virtual environments when there might be frequent updates of the environment, e.g., e-commerce platforms might want to frequently adjust the website design to enhance user intent inference.

We propose a framework for *GRD* with general agent behavior to address these limitations. To relax the optimality assumption, we explicitly incorporate models of agent behavior into the optimization framework. To tackle the computational challenges, our approach leverages data-driven methods for goal recognition design. The core idea is to build a machine learning oracle that predicts the difficulty of goal recognition (e.g., *wcd*) given a decision-making environment and an agent's behavioral model. This oracle is trained on datasets generated from simulations, enabling efficient evaluation of *wcd* for any given environment and agent model. Such an approach significantly accelerates the evaluation process during optimization. Once the machine learning oracle is established, we apply a general gradient-based optimization method to minimize *wcd*, using Lagrangian relaxation to handle constraints on environment modifications. Beyond addressing the two main limitations, this optimization procedure also supports more general forms of objectives and constraints that existing approaches in the literature cannot accommodate.

To evaluate our framework, we conducted a series of simulations and human-subject experiments. We start with simulations in the standard setup in the literature, with the grid world environment and optimal agent assumption. We show that our approach outperforms existing baselines in reducing worst-case distinctiveness (*wcd*) and achieves considerably better run-time efficiency. We then conducted additional simulations to demonstrate that our approach can generalize to settings that existing methods cannot address, including scenarios involving flexible budget constraints, more complex environments, and suboptimal agent behavior. Lastly, we have conducted human-subject studies demonstrating that our method can be leveraged to design environments that facilitate efficient goal recognition from real-world human decision-makers. The results highlight the potential of our approach to enable more efficient human-AI collaboration.

**Contributions.** The main contributions of this work can be summarized as follows.

- We propose a data-driven optimization framework that accommodates general agent behavior and improves the efficiency of goal recognition design, addressing two main limitations in the existing literature while enabling a more flexible optimization procedure. The framework consists of a predictive model that estimates the *wcd* for a given environment and a model of agent behavior. We then apply a gradient-based optimization method to perform goal recognition design. The framework is runtime efficient, capable of incorporating general agent behavior, and adaptable to different design spaces and constraints.

- Through extensive simulations, we show that our framework not only outperforms existing approaches in goal recognition design, both in reducing worst-case distinctiveness *wcd* and improving run-time efficiency in standard settings, but also adapts to scenarios that existing methods cannot handle, including general optimization criteria, complex environments, and suboptimal agent behavior.

- Through human-subject experiments, we demonstrate that our approach adapts to settings in which decision-making agents are real-world humans. To the best of our knowledge, our work is the first to evaluate the *environment design* for goal recognition with human-subject experiments.

## 2 Related Work

Our work contributes to the field of human–AI collaboration. Recent research shows that optimizing AI alone is insufficient to maximize the performance of human–AI teams (Bansal et al., 2021; Carroll et al., 2019; Yu et al., 2024), motivating efforts to understand and model human behavior in interactions with AI systems (Choudhury et al., 2019; Shah et al., 2019; Kwon et al., 2020; Chan et al., 2021; Reddy et al., 2021; Narayanan et al., 2022a;b; 2023; Treiman et al., 2023; 2024; 2025). To build truly collaborative AI agents, they must also anticipate the intentions and goals of their human counterparts. This challenge lies at the core of goal recognition research (Kautz, 1991; Ramírez & Geffner, 2010; Hong, 2001; Sukthankar et al., 2014; Pereira et al., 2017; Masters et al., 2021). Our work focuses on goal recognition design, an extension of goal recognition that incorporates environment modifications to make goals easier to recognize.

Goal recognition design ($GRD$) was formulated by Keren et al. (2014). Since this seminal work, numerous research efforts have extended to accommodate stochastic environments (Wayllace et al., 2017; Wayllace & Yeoh, 2022), varying levels of observability (Keren et al., 2016; Wayllace et al., 2020), and diverse design spaces (Mirsky et al., 2019). The studies most closely aligned with our approach focus on suboptimal agents (Keren et al., 2015; Wayllace & Yeoh, 2022). However, these studies characterize suboptimality by limiting deviations from the optimal policy, a method that may not adequately capture the behavior of human agents, who frequently deviate systematically from optimal decision-making. Moreover, most existing work requires evaluating goal recognition difficulty at run time across a large number of environmental modifications, which significantly limits scalability. Our work distinguishes itself by broadening $GRD$ to incorporate general models of agent behavior and by implementing a data-driven optimization approach.

There has been effort to accelerate $GRD$ by using heuristic approaches that avoid the prohibitively large search space of possible modifications. In their seminal work, Keren et al. (2014) proposed the pruned-reduce method, which restricts modifications to action removal and leverages a key insight: blocking actions do not introduce new goal-directed paths. This alleviates the computational burden of exhaustive search. More recent work by Pozanco et al. (2024) accelerates $GRD$ by using a plan library from a top-quality planner (Katz et al., 2020) to guide initial heuristic exploration. Similarly, Au (2024) expanded the design space with block-level modifications—simultaneous, interconnected changes—to improve efficiency. While these approaches make notable strides, they still rely on exhaustive search and repeated computation of complex metrics, which remains the primary computational bottleneck. In contrast, our method adopts a data-driven optimization framework. Rather than evaluating the metric ($wcd$) at each step of the optimization process, we train predictive models to approximate it directly, enabling rapid evaluation and comparison of design candidates. Moreover, the differentiability of the predictive model allows for gradient-based optimization in place of exhaustive search. Together, these advances significantly reduce computational overhead and open new possibilities for scaling $GRD$ to more complex settings.

Our methodology aligns with recent advances in data-driven optimization for mechanism design (Dütting et al., 2019; Golowich et al., 2018; Curry et al., 2020; Kuo et al., 2020; Rahme et al., 2020; Peri et al., 2021; Cornelisse et al., 2022; Yu et al., 2023), which leverage machine learning to accelerate otherwise intractable tasks. It also aligns with recent research that incorporates human models into learning and optimization processes (Ho et al., 2016; Evans et al., 2016; Shah et al., 2019; Tang & Ho, 2019; 2021; Carroll et al., 2019; Tang et al., 2021a;b; Yu & Ho, 2022; Feng et al., 2024; Kasumba et al., 2025). While we share the high-level motivation, our problem formulation and methodological choices offer novel contributions and advance the study of goal recognition design through a principled integration of predictive modeling and optimization.

## 3 Problem Formulation and Methods

### 3.1 Problem Setting

**Decision-making environment.** We define the decision-making environment as a Markov decision process (MDP), represented by $W = \langle S, A, P, R \rangle$. Here, $S$ denotes the set of states, $A$ represents the set of agent actions, $P(s'|s,a)$ is the transition probability from state $s$ to state $s'$ upon taking action $a \in A$, and $R(s,a,s')$ represents the bounded reward received after taking action $a$ in state $s$ and reaching state $s'$. To

emphasize the goal recognition aspects of the problem, we introduce a set of goal states $G \subseteq S$. These goal states are terminal; that is, $P(g|g, a) = 1 \ \forall g \in G, a \in A$.

**Models of behavioral agents.** We represent the agent's decision-making policy in a general form $\Pi : S \times T \to A$. Specifically, for an agent with a decision-making policy $\pi \in \Pi$, the agent will execute the action $\pi(s, t)$ in state $s$ at time $t$. The agent is conceptualized as a planner $H : W \to \Pi$, where the input is an environment $w \in W$, and the output is a policy $\pi \in \Pi$. To illustrate our formulation, consider an agent parameterized by a time-variant discounting function $d(t)$. The standard optimal agent model corresponds to a fixed discounting factor $\gamma \in (0, 1]$ with $d(t) = \gamma^t$. We can also represent an agent with present bias (O'Donoghue & Rabin, 1999) by adopting a hyperbolic discounting factor $d(t) = \frac{1}{1+kt}$, where $k > 0$. Note that our approach not only accounts for standard analytical closed-form expression of agent behavior. It can also account for scenarios where an agent policy $\pi$ is a machine learning model, i.e., a neural network trained on human behavioral data.

**Worst-case distinctiveness ($wcd$).** To evaluate the difficulty of goal recognition, we follow the standard literature and focus on worst-case distinctiveness ($wcd$) (Keren et al., 2014), defined as the maximum number of steps an agent can take that are consistent with multiple goals before its true goal becomes distinguishable. In other words, it measures the number of initial steps that overlap across trajectories toward different goals. To compute the $wcd$ for a given agent $h \in H$ in an environment $w \in W$, we evaluate the path for the agent to each goal and compute the number of actions from the initial state that are identical for each goal. While we focus on $wcd$ in this work, our method is agnostic to the specific metric used, as long as the metric is computable, a predictive model can be trained to approximate it and enable efficient evaluation.

## 3.2 Goal Recognition Design (GRD) Formulation

We formalize the $GRD$ problem with general behavioral agents. Given an environment $w$ and an agent with a behavior model $h$, we denote the worst-case distinctiveness of environment $w$ for agent $h$ as $wcd(w, h)$. Each type of modification $1 \leq i \leq N$ will incur a cost $c_i(w, w')$ that must fall in budget constraint $B_i$. The objective of the $GRD$ problem is to alter the environment from $w$ to $w'$ in a way that minimizes $wcd(w', h)$, while satisfying the constraint that the cost of the modifications does not exceed the budget.

$$\begin{aligned} \underset{w'}{\text{minimize}} \quad & wcd(w', h) \\ \text{subject to} \quad & c_i(w, w') \leq B_i, \forall 1 \leq i \leq N \end{aligned} \tag{1}$$

## 3.3 Our Proposed Method

Existing approaches to $GRD$ require evaluating $wcd$ for a large number of potential environment modifications in order to identify the optimal ones. Since computing $wcd$ involves evaluating the agent's policy across multiple goals, it introduces significant computational overhead and relies on strong assumptions about agent behavior. To address this challenge, we propose leveraging machine learning to expedite runtime computation and explicitly account for general agent models. The main idea, illustrated in Figure 1, is to first train a machine learning model that predicts $wcd$ for any given pair of decision-making environment $w$ and agent behavioral model $h$. Once this predictive model is trained, we exploit its differentiability to develop an optimization framework that applies gradient-based methods to the Lagrangian relaxation of the constrained optimization problem defined in Equation 1. This approach not only addresses the two key limitations—computational challenges and restrictive assumptions about agent behavior—but also provides flexibility for accommodating various forms of optimization objectives and constraints.

**Predictive model for $wcd$.** To build the predictive model for $wcd$, we curate a training dataset through simulations. For an environment $w$ and agent behavioral model $h$, we can obtain $wcd(w, h)$ by solving for the agent's actions towards different goals. After collecting a training dataset, we train the predictive model using a convolutional neural network. The implementation details are in Section 5.1.1 and the appendix.

**Optimization procedure.** After obtaining the predictive model, we develop a gradient-based optimization framework. This framework generalizes the existing literature in $GRD$ where the space of modifications

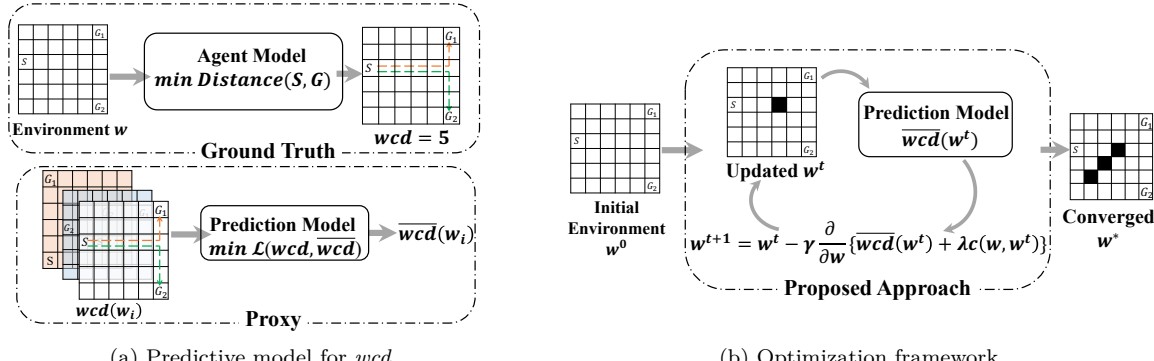

(a) Predictive model for *wcd*.

(b) Optimization framework.

Figure 1: We first train a *wcd* predictor from simulated data. We then perform gradient-based optimization that leverages the predictor to identify environment modifications that minimize *wcd* with a given agent behavior model.

is limited (e.g., usually limits to blocking an action in MDP) (Keren et al., 2021). The first step of the optimization procedure involves transforming the constrained optimization problem in equation 1 into an unconstrained optimization problem using Lagrangian relaxation:

$$\mathcal{L} = wcd(w', h) + \sum_i \lambda_i(c_i(w, w') - B_i).$$

We then perform gradient descent on the relaxed Lagrangian with respect to $w$. As environment modifications are often discrete (e.g., to block a cell in the grid world), we apply a discrete gradient descent procedure. Specifically, at each descent step, we obtain a gradient, which is a vector indicating the suggested change magnitude for each element (such as a cell in the grid world). We then select the element with the gradient value of the largest magnitude and make the corresponding change. Note that some suggested modifications may be invalid; for instance, we cannot block an already blocked cell in the grid world. In such cases, we proceed to the element with the next highest gradient value, continuing this process until a valid modification is made. This modification procedure is repeated until the gradient descent converges. Note that with this Lagrangian relaxation, while we cannot directly set the budget, based on duality, a larger Lagrangian multiplier $\lambda$ corresponds to a smaller budget $B$ in the original constrained optimization formulation. In practice, designer can choose to vary the size of the Lagrangian multiplier to modulate the budget. In our experiments, we choose varying Lagrangian multipliers and record the realized costs of the modifications.

## 4 Experiment Setup

### 4.1 Benchmark Domains

We utilize two benchmark domains. The first is the standard basic grid world environment, commonly used in the *GRD* literature. The second is the Overcooked-AI environment (Carroll et al., 2019), a more complex environment with a richer set of environment modifications. This environment is particularly relevant to the downstream implications of our work, namely in supporting human-AI collaboration.

#### 4.1.1 Benchmark Domain: Grid World

In the grid world domain, agents navigate a grid with several potential goals. Consider an example in Figure 2a: an agent starts at point 'S' and aims for one of the goals marked 'G'. Spaces marked 'X' are blocked. In this example, the worst-case distinctiveness (*wcd*) for an optimal agent is 0, as the shortest paths to the two goals do not overlap, meaning the agent's intended goal is revealed on the very first move. Our experiments primarily focus on grid world environments with two goals for simplicity. However, our approach extends naturally to settings with more than two goals by computing the relevant *GRD* metric and training a corresponding predictive oracle.

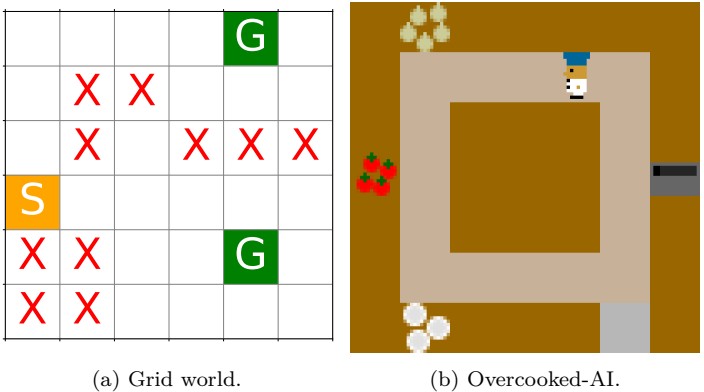

(a) Grid world.  (b) Overcooked-AI.

Figure 2: The benchmark environments. The left one is a grid world. The agent starts at a position marked 'S' and aims to reach one of the goal positions labeled 'G'. The agent must navigate through the grid, avoiding blocked cells marked with 'x'. The right one is an Overcooked-AI setting, where the agent's objective is to pick up ingredients and complete their target recipe, which constitutes their goal.

**Design space of environment modifications.** In the context of *GRD* in grid world, the space of environment modification is often limited to adding blocks to spaces in the literature (Keren et al., 2014), also called *action removal*. In our work, we broaden the design space to also consider the *unblocking* of existing blocked spaces as potential modifications.

### 4.1.2 Benchmark Domain: Overcooked-AI

We also conduct our evaluations in a more complex domain: Overcooked-AI.[1] This environment is based on the popular game Overcooked, where players (agents) collaborate to prepare and deliver specific recipes. Since the goal of our approach is to enable efficient inference of agent goals, we focus on the simplified setting with a single agent. The environment is represented as a grid (see Figure 2b), with each cell indicating the object located there. Objects may include a counter, tomato, onion, pot, dish, serving point, or empty space. The agent can only occupy empty spaces and cannot step on any other object. It can carry movable items, i.e., onions, tomatoes, and dishes, and place them on other non-space objects such as counters, serving points, or pots. To navigate the environment, the agent can move left, right, up, or down and maintains an orientation aligned with its most recent movement direction.

**Goal recognition in Overcooked-AI.** The agent's goal in Overcooked-AI is defined by the recipe it needs to complete. For example, to prepare a soup with one onion and two tomatoes, the agent must collect the ingredients, place them in a pot, and cook them. In the context of goal recognition, the objective is to infer which recipe the agent is pursuing based on its behavior. For simplicity, our experiments focus on scenarios with only two possible goals or recipes.

**Design space of environment modifications.** A primary challenge in the Overcooked-AI environment is its considerably larger design space for environment modifications. The design space includes changing the position of any object in the environment, subject to the constraint that the modification remains valid. This means that a change cannot result in a new environment design where any of the objects is unreachable by the agent. The objective of conducting experiments in this domain is to assess whether our approach can address more complex domains.

### 4.2 Baselines

We compare our approach against four baselines that collectively represent the major algorithmic directions in *GRD*: (Optimal) exhaustive search, heuristic-enhanced search, and greedy-based approaches. For heuristic-enhanced search, we implement the Pruned-Reduce method proposed by Keren et al. (2014) as a

---

[1]https://github.com/HumanCompatibleAI/overcooked_ai

representative example. For greedy-based approaches, we implement methods that use either the true or predicted *wcd* values to prioritize locally beneficial modifications.

- **Exhaustive search**: This is the brute-force approach that evaluates *wcd* for all the environments on the search path until the minimum possible *wcd* is found. It is guaranteed to achieve minimum *wcd*. However, given the computational overhead, this approach is not applicable in most situations.
- **Pruned-Reduce** (Keren et al., 2014): This baseline is specifically designed for settings where modifications are limited to action removal such as blocking a cell in grid world. It speeds up the exhaustive search and retains the optimal property. However, its scalability is still limited.
- **Greedy search using true *wcd***: This greedy search baseline finds the single environment modification that leads to the maximum reduction of *wcd* at each iteration. This approach requires to evaluate the *wcd* for all possible single environment modifications at each iteration.
- **Greedy search using predicted *wcd***: In addition to greedy search using true *wcd*, we leverage our predictive model for *wcd* and design another greedy baseline. This baseline finds the single environment modification that leads to the maximum reduction of predicted *wcd* at each iteration.

### 4.3 Models of Agent Behavior

In our experiments, to demonstrate that our approach works for different models of agent behavior, We have examined three types of agent behavior.

- **Optimal agent behavior**. The first one is the standard optimal agent behavior. Conducting experiments with optimal agent behavior enables us to compare our approach with standard approaches in the literature, which are often developed under the optimal agent assumption.
- **Parameterized suboptimal agent behavior**. We consider a generalized behavior model parameterized by $d(t)$. The agent's objective is to optimize a time discounted reward with the discounting factor for $t$ steps in the future being $d(t)$. The standard optimal agent model corresponds to a fixed discounting factor $\gamma \in (0, 1]$ with $d(t) = \gamma^t$. We can also represent an agent with present bias (O'Donoghue & Rabin, 1999) by adopting a hyperbolic discounting factor $d(t) = \frac{1}{1+kt}$, where $k > 0$.
- **Data-driven agent behavior.** We also address settings where the model of agent behavior is a machine learning model trained on human behavioral data.

## 5 Experiments

### 5.1 Simulations

In our simulations, we first evaluate how our approach compares with existing methods in standard setups commonly found in the literature. We aim to show that our method matches or exceeds existing approaches in reducing *wcd* while achieving significantly better time efficiency. We then demonstrate the generalizability of our approach by applying it to scenarios that state-of-the-art methods cannot address, including those with dynamic budget constraints, more complex environments, and suboptimal agent behavior.

**Standard setting.** We start with settings within the grid world domain, where modifications are limited to blocking cells and agent behavior is assumed to be optimal. This is the standard setting for the majority of goal recognition design studies, as highlighted in Table 1 of the survey by Keren et al. (2021). The objective of this set of simulations is to enable comparisons with state-of-the-art methods.

In our experiments, the initial environment is generated randomly: we first randomly select the number of blocked cells from the range $[0, 12]$, followed by randomly allocating the blocked cells. We also randomly determine the starting grid and two goal grids. Environments where the goals are not reachable from the starting grid are filtered out. We randomly generate at least 500 environments and compare the average performance of our approach with that of baseline methods. More details of data generation and *wcd* predictive model training are provided in Appendix A.1 and C. During optimization for our approach, the budget cannot be directly specified, so we explored over a range of Lagrangian parameters on a log scale

between 0 and 10 (detailed in Appendix subsection B.2) and record the resulting realized budgets. We use the realized budget in our experimental figures because there is a monotonic relationship: larger Lagrangian parameters correspond to stricter penalties, generally producing lower realized budgets. This allows for a clearer comparison between methods that set budgets directly and our approach, which controls budgets indirectly through Lagrangian parameters.

We begin our experiments with a $6 \times 6$ grid world. In this simplest setting, our approach and all baselines achieve similar performance in reducing *wcd*, as shown in Figure 3a. However, our method demonstrates a significant run-time advantage over exhaustive search. In particular, our approach requires only 0.2 seconds, compared to approximately 2 seconds for exhaustive search.

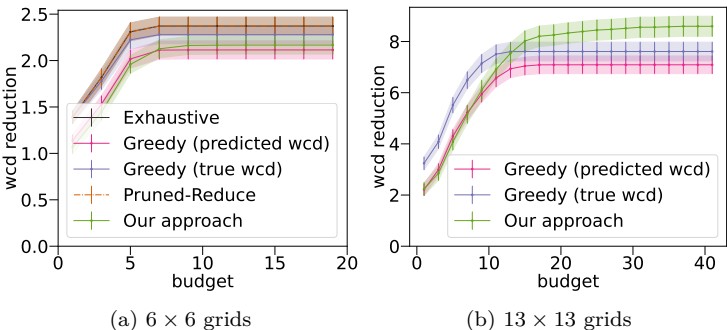

(a) $6 \times 6$ grids      (b) $13 \times 13$ grids

Figure 3: *wcd* reduction in a grid world when only blocking modifications are allowed. Exhaustive search and Pruned-Reduce are not included in (b) because they take more than an hour to compute for a single environment.

We then scale up the environment to a $13 \times 13$ grid, where both exhaustive search and the Pruned-Reduce method failed to complete within an hour for a single instance and were therefore excluded from the baselines. As shown in Figure 3b, our approach outperforms the greedy baselines in reducing *wcd*, due to its budget-aware optimization procedure that systematically accounts for the long-term impact of each modification—making it less prone to shortsighted or myopic decisions. Additionally, our method is approximately three times faster than the greedy baselines for large budgets (with further runtime comparisons discussed later).

**Flexible budget constraints.** In the literature, most works focus on a single type of environment modification (e.g., blocking a cell). Given the flexibility of our optimization framework, we extend our approach to include 'unblocking' blocked cells as a possible environment update. In our simulations, we examine two common cases. In "shared budget", the total number of blocking and unblocking actions is bounded by a given shared budget. In "individual budget", we limit the number of blocking and unblocking actions separately. Specifically, we allow the number of blocking actions

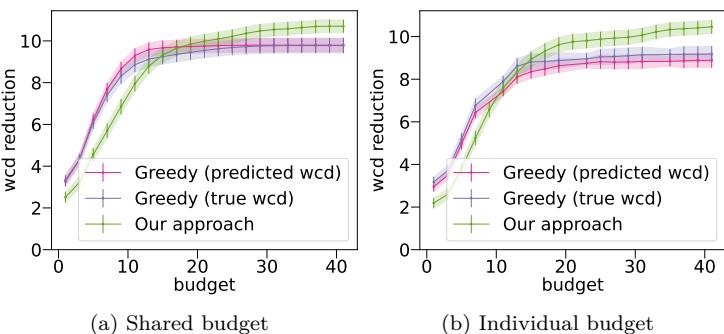

(a) Shared budget      (b) Individual budget

Figure 4: *wcd* reduction in settings with two types of modifications. We only included greedy as the state-of-the-art baselines - Exhaustive search and Pruned-Reduce are not applicable in these settings.

to be 5 times the number of unblocking actions (rounded to the nearest integer). Given that there are no established baselines (Pruned-Reduce is limited to blocking only modifications, and Exhaustive Search does not scale) for this setting in the literature, we compare our results against greedy baselines.

The results are shown in Figure 4. Overall, they demonstrate that our approach is effective at reducing *wcd* even when existing state-of-the-art methods are not applicable. In addition, our gradient-based optimization outperforms the greedy approach except for small-budget cases[2]. This is because our method considers how to distribute the budget over time, rather than selecting the highest-impact modification at each step without considering future opportunities.

---

[2]Note that by definition of greedy approaches, the greedy approach using the true *wcd* is optimal when the budget is 1, since only one modification can be made. As a result, greedy methods generally perform well when the budget is very small, but our approach starts to outperform greedy with larger budgets.

**Complex domain and suboptimal agent behavior.** We consider two additional extensions, where existing approaches in the literature do not apply, to demonstrate the flexibility of our framework. In the first, we evaluate our approach in a more complex problem domain: Overcooked-AI. In the second, we return to the grid world but include scenarios with suboptimal agent behavior. Both extensions utilize a grid size of $6 \times 6$. For the Overcooked-AI environment, we assume optimal agent behavior and aim to explore how our approach adapts to a much richer space of environment modifications. For the suboptimal human behavior, we employ the model described in Section 4.3, utilizing a hyperbolic discounting factor $d(t) = \frac{1}{1+kt}$ and set $k = 8$. For both extensions, standard approaches such as exhaustive search and Prune-Reduce are either too slow or not applicable. Therefore, we compare our results with the greedy baselines.

The results for both extensions, as illustrated in Figures 5a and 5b, demonstrate that our approach adapts well to both settings. Our approach demonstrates a more significant advantage over baselines in Overcooked-AI than in grid world settings. The increased complexity and larger modification space in Overcooked-AI cause greedy methods to converge to short-sighted solutions, underscoring the superior scalability of our method. Regarding runtime, our approach is again several orders of magnitude faster than the greedy method with true *wcd* and at least 3 times faster than the greedy method using predicted *wcd*.

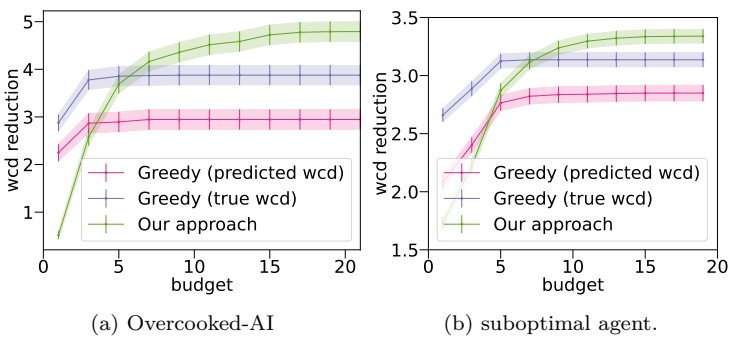

(a) Overcooked-AI  (b) suboptimal agent.

Figure 5: Performance in other settings: (a) Overcooked-AI environment with optimal agent behavior, and (b) Grid world ($6 \times 6$) with suboptimal agent behavior, demonstrating the method's adaptability to non-optimal decision-making.

**Running time comparisons.** We have demonstrated that our approach achieves greater *wcd* reductions compared to all baselines. Another key benefit of our method is its significantly improved runtime efficiency relative to standard approaches. This efficiency comes from (1) using a machine learning oracle to predict *wcd* instead of evaluating the true *wcd* at each time step, and (2) applying gradient-based optimization rather than the (heuristic-enhanced) exhaustive search methods used in the literature. As shown in our earlier simulations, exhaustive search methods scale poorly. In the 13x13 environment, a single instance of *GRD* takes more than one hour, while our approach completes in about one second. Therefore, we focus on the improvements due to the ML oracle in the following comparisons.

Table 1: Time (seconds) comparisons when using true *wcd* and predicted *wcd*. $\pm$ denotes standard error.

| Setting | True *wcd* | Predicted *wcd* | |
|---|---|---|---|
| | greedy | greedy | our approach |
| Standard (small) | 0.11±0.01 | 0.47±0.0 | 0.5±0.0 |
| Standard (large) | 4.95±0.99 | 2.46±0.49 | 1.11±0.11 |
| Suboptimal behavior | 183±4 | 0.59±0.02 | 0.71±0.02 |
| Complex domain | 531±15 | 0.41±0.01 | 0.8±0.2 |

Table 1 shows the time required to make modifications under a fixed budget across different settings. The results indicate our methods are generally faster and scale better in complex environments. For instance, in the complex domain, computing the true *wcd* takes 531 seconds for *GRD*, whereas our approach takes less than 1 second. Additional discussion on the time improvements is provided in Appendix A.2.

### 5.1.1 Ablation Study

We conducted an ablation study to better understand the specific contributions of our proposed approach to goal recognition design (*GRD*), particularly its ability to reduce *wcd*. The study isolates two key components

of our method: the accuracy of the predictive model used to estimate *wcd* and the effectiveness of the optimization procedure in modifying the environment to improve goal recognition.

We evaluated several predictive models trained to estimate *wcd* and tested them in combination with different optimization strategies. These are compared against a baseline that applies a greedy search using the true *wcd* without relying on a predictive model. All experiments were conducted under three environment modification settings: the standard setting and two variants with flexible budget constraints (individual and shared budgets). Results for environments with grid size 13 are shown in Table 2, with additional results for grid size 6 provided in the appendix.

**Impact of predictive model choice.** We trained several predictive models, including Convolutional Neural Networks (CNNs), linear models, and Transformers, using the Adam optimizer (Kingma & Ba, 2015) and selected the configurations through validation. CNNs consistently achieved the lowest prediction error, and crucially, enabled more effective environment modifications when combined with either greedy or gradient-based optimization. By contrast, linear models and Transformers performed worse in predicting *wcd* and led to less effective downstream performance. This highlights the importance of predictive accuracy, where the quality of the predicted *wcd* directly influences the success of environment design.

**Impact of optimization approach.** We also analyzed the role of the optimization procedure given a fixed predictive model, using (i) greedy optimization using predicted *wcd*, and (ii) our proposed gradient-based optimization approach. Greedy optimization using predicted *wcd* can offer runtime benefits over using true *wcd*, but its effectiveness varies with prediction quality. In cases where the predictor was less accurate, the greedy method often failed to produce significant improvements in *wcd*. Our proposed gradient-based approach consistently improved upon the greedy method. In particular, when combined with the most accurate CNN predictor (100K samples at a learning rate of 0.001), our method achieved the lowest *wcd* across all three modification settings, significantly beating the greedy baseline using true *wcd* as well.

The best-performing configuration, CNN (100K, 0.001) combined with our gradient-based optimization, achieved the greatest reduction in *wcd* and outperformed all baseline approaches. This supports our core claim: accurate prediction of *wcd* paired with an effective optimization strategy, as adopted in our approach, is critical to designing environments that enhance goal recognition.

Table 2: Ablation results showing the impact of different predictive models and optimization methods on *wcd* reduction across three environment settings as described in subsection 5.1. Each row represents a predictive model trained with a specified dataset size, learning rate, and associated training/validation loss. Due to poor performance, only the best Linear and Transformer configurations are shown. For each model, we compare two optimization methods: Greedy using predicted *wcd* and our proposed approach, along with a baseline using greedy with true *wcd*. Bolded results indicate the best performance per setting. Results are for grid size 13; grid size 6 results are in Appendix B.3

| Predictive Model | Loss (mse) | | Optimization Approach | *wcd* Reduction | | |
| --- | --- | --- | --- | --- | --- | --- |
| | Train | Validation | | Standard | Individual | Shared |
| Baseline (NA) | NA | NA | Greedy (true *wcd*) | 7.6±0.4 | 9.2±0.4 | 9.8±0.4 |
| Linear (100K, 0.1) | 14 | 15 | Greedy (predicted *wcd*) | 1.9±0.3 | 1.2±0.2 | 1.2±0.2 |
| | | | Our Approach | 2.1±0.2 | 4.8±0.3 | 5.4±0.3 |
| Transformer (100K, 0.1) | 20 | 20 | Greedy (predicted *wcd*) | 0.0±0.0 | 0.1±0.1 | -0.6±0.1 |
| | | | Our Approach | 1.5±0.2 | 4.2±0.2 | 4.5±0.3 |
| CNN (1K, 0.001) | 0.6 | 13 | Greedy (predicted *wcd*) | 1.9±0.2 | 0.8±0.2 | 1.0±0.2 |
| | | | Our Approach | 1.9±0.2 | 3.9±0.3 | 4.0±0.3 |
| CNN (10K, 0.001) | 0.2 | 4 | Greedy (predicted *wcd*) | 4.3±0.3 | 3.8±0.3 | 4.1±0.3 |
| | | | Our Approach | 5.1±0.3 | 8.5±0.3 | 9.3±0.3 |
| CNN (100K, 0.001) | 0.1 | 0.2 | Greedy (predicted *wcd*) | 7.1±0.4 | 8.9±0.3 | 9.8±0.3 |
| | | | Our Approach | **8.6±0.4** | **10.4±0.3** | **10.8±0.3** |

## 5.2 Real-World Human-Subject Experiments

In our simulations, we demonstrate that our approach consistently outperforms baseline methods in terms of *wcd* reduction and offers greater efficiency in runtime. To assess the applicability of our approach when humans are the decision-makers, we conducted two sets of human-subject experiments. In the first, we aim to collect human behavioral data in our decision-making environments. Utilizing this data, we employ imitation learning to develop a model that accurately represents human behavior. This model is then integrated as the agent behavior model within our approach. In our second experiment, we aim to evaluate whether our approach indeed leads to environments that facilitate more effective goal recognition by human decision-makers. These experiments are approved by the Institutional Review Board (IRB) of our institution.

### 5.2.1 Experiment 1: Collection of Human Behavioral Data

We recruited 200 Amazon Mechanical Turk workers, paying each \$1.00 for an average task time of 3.64 minutes, equivalent to an hourly rate of about \$16. Each participant is asked to play 15 navigation games in a $6 \times 6$ grid world, navigating from a start position to a designated goal in each game. At each time step, they could choose to move in one of four directions: Up, Down, Right, or Left. A game concluded when the participant reached the goal. The environments for these games were generated similarly to our simulations, with start positions, goal positions, and block positions all being randomly determined. Our objective with this setup was to leverage the collected data to develop a data-driven model of human behavior.

**Learning models of human behavior.** The collected human data were split into training (160 workers, 70,000 user decisions), validation, and testing sets (20 workers, 8,800 decisions each). A 4-layer Multilayer Perceptron (MLP) was trained, with the environment layout as input to predict the next human action. We fine-tuned hyperparameters and chose the model architecture using validation. We compared the performance of our learned model against one that assumes optimal agent behavior i.e., taking the shortest path to the goal. Table 3 presents both models' accuracies, showing that human behavior significantly deviates from optimality. This deviation underscores the importance of incorporating a realistic model of human behavior in *GRD*, particularly when humans are the decision-makers.

Table 3: Prediction accuracy of human behavior assuming optimal behavior versus using a data-driven model.

| Model | Train | Val. | Test |
|---|---|---|---|
| Assuming optimal behavior | 0.7266 | 0.6964 | 0.7131 |
| Data-driven model | 0.9189 | 0.8136 | 0.8422 |

### 5.2.2 Experiment 2: Evaluating Goal Recognition Design

We next evaluate our approach that incorporates the data-driven model of human behavior from Experiment 1. We randomly generate 30 initial environments using the same setup as in simulations. These environments are then updated according to four different methods, all operating within a modification budget of 20.

- Original: No updates to the environment.
- Greedy: Greedy baseline using predicted *wcd* from the data-driven human behavior model.
- Proposed (opt-bhvr): Our proposed approach when assuming the agent is following optimal behavior (i.e., picking one of the shortest paths towards the goal).
- Proposed (data-driven): Our proposed approach using predicted *wcd* from the data-driven model.

We recruited 200 workers from Amazon Mechanical Turk. Each worker was randomly assigned to one of the four treatments above, with the distinction between treatments being the environments presented to them. Workers were randomly assigned a goal for each environment and tasked with navigating to reach it. We utilize the collected data to evaluate *GRD* design approaches.

**Comparing empirical overlapping actions.** To evaluate the effectiveness of each *GRD* approach, we first measure the empirical overlapping actions toward each of the two goals. Specifically, each recruited worker is exposed to all 30 environments in their assigned treatment and is instructed to reach one of two randomly selected goals. For each treatment, we compute the number of overlapping actions for every pair of workers assigned to different goals within the same environment. This yields a distribution of overlapping actions for each treatment. This metric reflects the difficulty of inferring an agent's goal and serves as an empirical

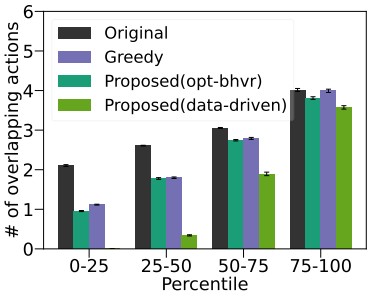
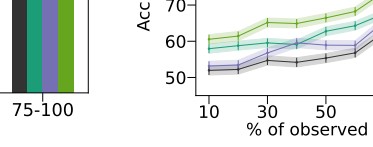

(a) Empirical overlapping actions (the lower the better).

(b) Bayesian goal inference (the higher the better).

Figure 6: Comparing different *GRD* approaches in our human subject experiments. Our approach coupled with data-driven models is shown to generate environments that enable the most effective goal recognition with real human decision-makers.

proxy for *wcd*. Figure 6a presents the percentiles of the number of overlapping actions across treatments. Notably, in the lowest 25%, environments generated by our approach show no overlapping actions, indicating that goal inference is relatively easy in these settings. Overall, when combined with data-driven models, our approach produces environments with statistically significantly fewer overlapping actions (paired t-test, p-value $< 0.005$), thereby facilitating more accurate identification of human goals.

**Comparing the accuracy of goal inference.** Instead of evaluating the effectiveness of our approaches based on *wcd* or its proxy, we next directly assess whether human goals can be accurately inferred from partial observations of their actions. To do this, we use an off-the-shelf Bayesian inference algorithm (Murphy, 2012) [3]. As the agent acts, we update the posterior belief using the likelihood derived from our data-driven behavior model. Specifically, for each worker, we reveal the first $k$ portion of their actions to the inference algorithm, which then attempts to infer the worker's goal. This allows us to compute the average inference accuracy. The results, shown in Figure 6b, demonstrate that our approach produces environments that make goal recognition easier.

## 6 Conclusion and Future Work

Effective human-AI collaboration hinges on the AI system's ability to infer human goals. In this work, we introduce a data-driven framework for goal recognition design (*GRD*) that optimizes environments to enhance goal disambiguation. Our method addresses two major limitations of prior approaches: computational inefficiency and the unrealistic assumption of fully optimal agent behavior. At the core of our framework is a learned oracle that predicts goal recognition difficulty, worst-case distinctiveness (*wcd*). By integrating this oracle into a gradient-based optimization process, we avoid expensive exact evaluations and unlock a scalable, flexible design procedure. This enables three core advantages: (1) improved scalability, (2) support for general, potentially suboptimal agent models, and (3) extensibility to diverse constraints and objectives.

Our empirical evaluation highlights the effectiveness and versatility of the approach. In both classical grid worlds and complex Overcooked-AI environments, our method consistently outperforms baseline techniques in reducing *wcd* while significantly lowering computational cost. Crucially, it remains effective under suboptimal agent behavior—demonstrating robustness where many existing methods break down. Human-subject experiments further validate the practical utility of our framework, showing that it generalizes beyond simulation and maintains its advantage even when agent models are learned from real human data. To our knowledge, this is the first work to evaluate *GRD* methods in a human-subject setting. Through ablation studies, we demonstrate that the predictive oracle plays a critical role in guiding effective environment optimization, with more accurate or expressive models leading to significantly better outcomes.

---

[3]We assume a uniform prior distribution over goals. As the agent takes action, we update the posterior belief over goals through Bayesian updates based on the likelihood probability of the data-driven behavior model prediction.

**Limitations and future work.** While our framework advances goal recognition design, several limitations remain. First, our approach requires a large amount of data to train accurate predictive oracles. Although simulations can be used to generate training data, data from novel and richer environments is often scarce and presents challenges for simulation. Future work could explore several methods to address this limitation, including leveraging transfer learning (Pan & Yang, 2009) by pretraining on simulated environments and fine-tuning on limited data from new environments, as well as using generative models to produce synthetic yet behaviorally plausible training data (Goodfellow et al., 2014; Xu et al., 2019). Active learning techniques that reduce the amount of data required during collection also offer a promising direction. While these strategies require further validation, we believe they represent potentially viable and interesting avenues for applying *GRD* in data-scarce settings.

Second, we assume fully observable environments and stationary human behavior in this work. To address these limitations, future work could train the predictive model on inputs derived from belief state representations, incorporating uncertainty in agent observations (Kaelbling et al., 1998) into the training process. In addition, the predictive model could be periodically fine-tuned using newly accumulated behavioral data, allowing it to adapt to changes in human decision-making over time.

Moreover, our investigation focuses on single-agent, discrete environments. To extend our framework to multi-agent settings, future work would need to model agent behavior in the presence of other agents, accounting for their interactions. To support continuous environments, the Lagrangian relaxation can be adapted to penalize changes using continuous distance functions such as norms, enabling gradient-based optimization without altering the underlying pipeline. Overall, the modularity of our framework enables future work to address these limitations in a targeted and flexible manner.

**Broader impact.** Finally, while our work aims to advance human–AI collaboration through goal recognition design, it is important to acknowledge the societal risks that may arise from its misuse.

First, designing environments that make human goals and intentions easier to infer could be misused. Systems capable of anticipating intentions might be deployed not to assist users but to manipulate or steer their behavior. Because system designers typically hold an advantageous position over decision makers within these environments, such capabilities can further amplify their influence over individuals. Second, since *GRD* relies on predictive models trained on human behavioral data, it raises privacy concerns. Even when raw data are not directly exposed, adversaries could analyze model outputs or behaviors to infer sensitive personal attributes, creating opportunities for profiling or surveillance. Third, by making goal inference more accurate and efficient, *GRD* could foster excessive reliance on AI predictions. Such overreliance risks undermining human autonomy and weakening critical judgment, especially if users begin to accept AI inferences uncritically. Taken together, these risks highlight how a technique intended to enhance collaboration could, under certain deployments, compromise user welfare and agency.

In light of these concerns, we discuss possible avenues for risk mitigation. On the technical side, privacy-preserving methods such as differential privacy could be integrated into the training of behavioral models, limiting the extent to which sensitive information is encoded and thereby reducing opportunities for exploitation. Moreover, incorporating uncertainty-aware methods into *GRD* could ensure that systems explicitly communicate the confidence of their inferences. Such mechanisms would help mitigate both privacy risks and overreliance by reminding users of the fallibility of AI systems. On the policy side, once the capabilities and risks of *GRD* are more fully understood, it will be important to weigh the potential benefits against the possible harms. This evaluation should guide the development of regulatory frameworks and ethical guidelines—for example, requiring that *GRD*-driven modifications not significantly reduce user autonomy relative to standard environments, and mandating transparency about the objectives and data underlying deployment. Establishing such safeguards will be essential to ensure that *GRD* is developed and applied responsibly, and that its benefits are realized without compromising broader societal values.

## Acknowledgments

This work is supported in part by a J.P. Morgan Faculty Research Award and a Global Incubator Seed Grant from the McDonnell International Scholars Academy.

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

# A    Experiment Details and Additional Results

In this section, we provide details of the experiments not included in the main paper due to space constraints. We also include and discuss additional experimental results.

## A.1    Implementation Details

This subsection describes the implementation of the predictive model. We trained the predictive models on a simulated dataset generated by solving *wcd* for randomly sampled environments with a specific agent behavior model. Several architectures were tested, including CNNs, Linear Models, Kernel Ridge Regression, and Transformers, with CNNs performing the best in predicting *wcd*. The models processes inputs of the shape $k \times N \times N$ inputs, where $k$ is the number of potential objects in an $N \times N$ grid.

The CNN architecture used across the paper employs a ResNet18 backbone with an adapted $k$-channel input layer, followed by three fully connected layers. Outputs are reduced to 32 and 16 dimensions with Leaky ReLU activations and dropout, culminating in a single non-negative scalar via ReLU activation.

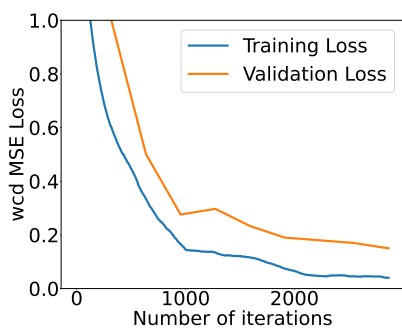

For the experiments in Section 5, we trained a CNN-based model on 100K data points, split into 80% training and 20% validation. The model achieved an MSE below 0.18 for both sets, with the validation set used for hyperparameter tuning. Figure 7 shows the training loss for the *wcd* predictive model for the grid world domain ($13 \times 13$) assuming optimal behavior. We used Adam optimizer and MSE loss and tested learning rates of 0.1, 0.01, 0.001, and 0.0001. A learning rate of 0.001 consistently produced the lowest validation error, enabling reliable *wcd* reduction across different setups.

Figure 7: Training loss and validation errors for the CNN predictive model for *wcd* with $13 \times 13$ grid world and optimal human behavior.

## A.2    More Results for Run Time Comparisons

Due to space constraints and also that the results align with one would expect, we do not include the run time details for different approaches in the main text. We include the results here for completeness. Overall, as shown in Figure 8, approaches that leverage the predictive model for *wcd* are orders of magnitude faster than methods that require to evaluate *wcd* during run-time. Note that the y-axis of the figure is in log scale so the difference is in at least two orders of magnitude.

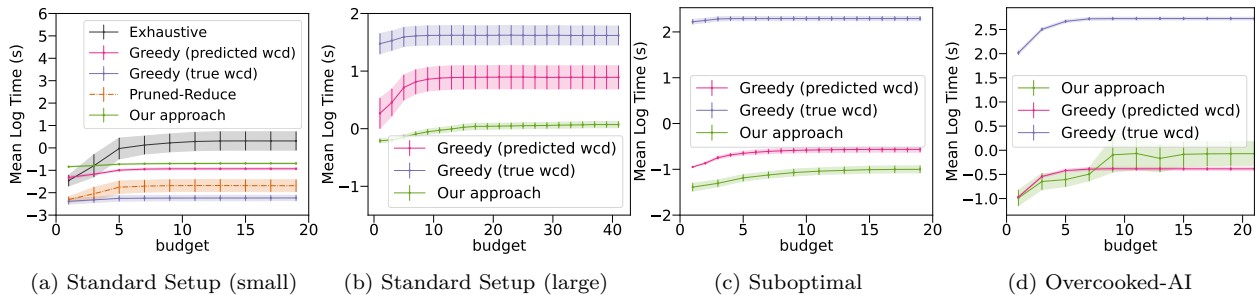

(a) Standard Setup (small)     (b) Standard Setup (large)     (c) Suboptimal     (d) Overcooked-AI

Figure 8: Run time taken by each method in the different experimental conditions. In the standard setup with a small grid size, all methods except exhaustive complete within less than half of a second but our approach scales better with more complex configurations.

# B   Additional Experiment Results

In this section, we report the additional experiment results that are not included in the main text due to space constraints. In Section 5.1 of the main paper, we provided details of our performance with a large grid size. In a smaller grid world, our approach achieves comparable performance to the baselines but it significantly outperforms them in larger grid sizes as shown in Figure 9.

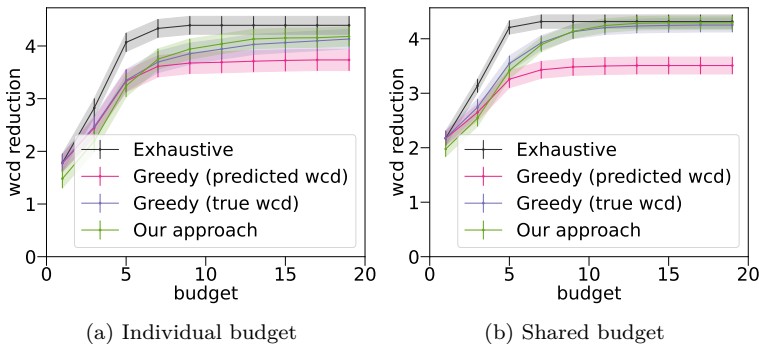



(a) Individual budget                    (b) Shared budget



Figure 9: *wcd* reduction in smaller ($6 \times 6$) gridworld with optimal agent behavior.

## B.1   Definition of Budget

In our evaluation, we compare the *wcd* reduction relative to the budget allocated for modifications. In the grid-world domain, the budget represents the number of changes made, which are limited to two types: blocking or unblocking cells. In contrast, the Overcooked-AI domain allows for a richer set of modifications due to its complexity. Here, modifications involve changing the positions of objects. A valid environment must include all specified objects, as detailed in Section 4.1.2. The budget in this domain is quantified as the total Manhattan distance moved by the objects between the original and modified environments.

## B.2   Langrange Parameter Space

In the main paper, we described how we relax the original constrained optimization problem by introducing a Lagrangian penalty parameter $\lambda$, which indirectly controls the budget through penalizing constraint violations. Since the budget cannot be specified directly, we instead vary $\lambda$ across a predefined range to explore different trade-offs between optimizing the objective and satisfying the budget constraint. Specifically, we sweep over a log-scaled set of values:, $\lambda \in \{0, 0.001, 0.002, 0.005, 0.007, 0.01, 0.02, 0.05, 0.07, 0.1, 0.2, 0.5, 0.7, 1.0, 2, 5, 7\}$. This range covers several orders of magnitude, allowing the optimization to balance constraint enforcement from very loose (small $\lambda$) to very strict (large $\lambda$) penalties. For all experiments, including those with a single budget constraint, we used this same set of $\lambda$ values. In scenarios involving two constraints, such as Blocking and Unblocking constraints, we applied this range independently to each constraint, effectively exploring the joint penalty space by combining these $\lambda$ values for each constraint.

## B.3   Ablation Analysis

In the main paper, we provided the ablation analysis for the optimal behavior model with a grid size of 13. Table 4 provides the same analysis for grid size 6. The results show mixed performance in this smaller environment, and our approach sometimes underperforms baselines. This reflects that our method's advantages emerge primarily in larger, more complex environments where traditional approaches become computationally intractable.

Table 4: Ablation results for grid world (size=6×6) showing the impact of different predictive models and optimization methods on *wcd* reduction across three environment modification settings, i.e, Standard Setting, Individual Budget, and Shared Budget as described in subsection 5.1. Each row represents a predictive model (e.g., CNN) trained with a specified dataset size, learning rate, and associated training/validation loss. Due to poor performance, only the best Linear and Transformer configurations are shown. For each model, we compare two optimization methods: Greedy using predicted *wcd* and our proposed Approach, along with a baseline using greedy optimization with true *wcd*.

| Predictive Model | Loss (mse) | | Optimization Approach | *wcd* Reduction | | |
|---|---|---|---|---|---|---|
| | Train | Validation | | Standard | Individual | Shared |
| Baseline 1 (NA) | NA | NA | Exhaustive Search (True *wcd*) | 2.4±0.1 | 4.5±0.2 | 4.3±0.1 |
| Baseline 2 (NA) | NA | NA | Greedy (True *wcd*) | 2.3±0.1 | 4.2±0.2 | 4.3±0.1 |
| Linear (100K, 0.1) | 1.9 | 2.1 | Greedy (Predicted *wcd*) | 1.1±0.1 | 1.6±0.1 | 0.0±0.0 |
| | | | Our Approach | 0.4±0.0 | 1.4±0.1 | 1.4±0.2 |
| Transformer (100K, 0.1) | 3.0 | 3.0 | Greedy (Predicted *wcd*) | 0.0±0.0 | 0.4±0.1 | 0.0±0.1 |
| | | | Our Approach | 0.0±0.0 | 0.1±0.1 | 0.4±0.1 |
| CNN (1K, 0.001) | 0.1 | 1.7 | Greedy (Predicted *wcd*) | 1.0±0.1 | 1.4±0.2 | 1.5±0.1 |
| | | | Our Approach | 1.0±0.1 | 2.9±0.2 | 2.8±0.2 |
| CNN (10K, 0.001) | 0.0 | 0.4 | Greedy (Predicted *wcd*) | 2.0±0.1 | 3.7±0.2 | 3.6±0.1 |
| | | | Our Approach | 1.9±0.1 | 4.1±0.2 | 4.0±0.1 |
| CNN (100K, 0.001) | 0.0 | 0.0 | Greedy (Predicted *wcd*) | 2.1±0.4 | 3.8±0.2 | 3.5±0.2 |
| | | | Our Approach | 2.2±0.1 | 4.3±0.2 | 4.3±0.1 |

## C   Details of Simulations

To generate our training and evaluation datasets, we randomly generated environments and kept valid environments, e.g., those in which the goals are reachable and the objects don't overlap. Below, we provide more details about environment generation for specified grid sizes for both overcooked and grid-world domains.

### C.1   Grid world

In a grid world, a valid environment includes a starting position, blocked cells, and two goal positions. The starting position is randomly placed in the first column, and the goal positions are randomly placed in the last two columns. The number of blocked positions in each grid is randomly selected from a range of 0 to 2× the grid width. We discard any assignments that make the goals unreachable. For experiments with suboptimal behavior, we also randomly assigned small subgoal rewards to unblocked cells that the agent would collect on its way to the goal state. The two goal states are assigned a large reward that is 10 times the largest subgoal reward.

### C.2   Overcooked-AI

In Overcooked-AI, a valid environment includes one pot, one tomato source, one onion source, one dish source, one serving point, no open spaces at the border, and any number of open spaces and blocked cells. All objects must be reachable from the agent's randomly assigned starting position, with the number and positions of blocked cells assigned randomly. The agent is randomly assigned one of two goals for each experiment, where both goals are randomly selected from the set of three possible recipes: three-tomato soup, three-onion soup, or mixed soup. Each goal has the same randomly assigned reward value. Suboptimal behavior is modeled by assigning small random subgoal rewards when adding ingredients to the pot.

## D   Details of Human-Subject Experiment

Lastly, we include more information about our human-subject experiments. In the human-subject experiment, each worker is asked to play a navigation game in 6 × 6 grid world environments. The task interface is shown in Figure 10.

Note that while each environment has two goals, we only show one goal (the goal of the worker) to the worker in our interface to simplify the presentations. The second goal is shown as a blocked cell in the interface, i.e the worker only navigates to the shown goal.

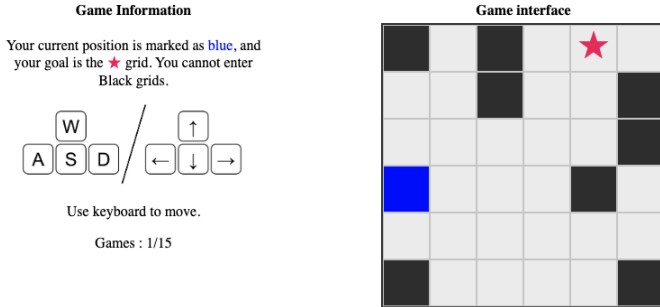

Figure 10: The interface of our human subject experiment.

We have recruited 200 workers from Amazon Mechanical Turk in total, and Table 5 contains the demographic information of the workers.

Table 5: Demographic information of the participants in our experiment.

| Group | Category | Number |
|---|---|---|
| Age | 20 to 29 | 84 |
| | 30 to 39 | 76 |
| | 40 to 49 | 26 |
| | 50 or older | 14 |
| Gender | Female | 89 |
| | Male | 110 |
| | Other | 1 |
| Race / Ethnicity | Caucasian | 175 |
| | Black or African-American | 8 |
| | American Indian/Alaskan Native | 3 |
| | Asian or Asian-American | 8 |
| | Spanish/Hispanic | 1 |
| | Other | 5 |
| Education | High school degree | 5 |
| | Some college credit, no degree | 5 |
| | Associate's degree | 4 |
| | Bachelor's degree | 135 |
| | Graduate's degree | 49 |
| | Other | 2 |

# E   Computation Details

All experiments were run on a computing cluster equipped with 40 CPU cores (Intel Xeon Gold 6148 @ 2.40GHz), a single NVIDIA Tesla V100 SXM2 GPU (32GB), and up to 80GB of memory. To ensure reproducibility, we worked within a clean and consistent software environment built around Python 3.10 and widely used scientific libraries. PyTorch was our main deep learning framework, with NumPy and pandas handling numerical computation and data processing. During model training and the discrete gradient descent optimization, we relied on PyTorch's autograd system to compute gradients automatically and efficiently.

