# OpenReview forum: "Goal Recognition Design for General Behavioral Agents using Machine Learning"
_TMLR — Accepted by TMLR_

### Review · Reviewer_dHQo · 2025-06-13

**Summary Of Contributions:**

This paper presents a data-driven framework for goal recognition design that improves scalability and accounts for general, including suboptimal, agent behavior. The authors train a predictive model to estimate the difficulty of goal recognition via worst-case distinctiveness, and use it in a gradient-based optimization process to modify environments efficiently. Their method outperforms existing approaches in both worst-case distinctiveness reduction and runtime, scales to complex settings, and handles flexible constraints. Human-subject experiments further demonstrate its effectiveness in enhancing goal recognition for real-world human decision-makers.

**Audience:**

Yes

**Broader Impact Concerns:**

No.

**Claims And Evidence:**

Yes

**Requested Changes:**

The authors may revise their manuscript according to the following questions:
* In the first paragraph of page 5 (optimization procedure), the authors proposed to solve the problem (1) via optimizing over the unconstrained Lagrangian relaxation. Since strong duality does not hold here and the dual problem is not solved, the obtained optimal point $w'$ is not globally optimal. While I do not think this should be an issue in practical applications, the authors mentioned on top of page 8 that "...as our optimization procedure accounts for the available budget and avoids getting trapped in locally optimal solutions". So I am wondering why this claim hold? Was this conclusion made from explaining the experiment data, or did I missed something?
* Based on the previous question, the budged constraints in the prolem (1) can (and are likely to be) violated if one simply optimizing the Lagrangian without carefully tuning the $\lambda$. So I am wondering is the budged really a hard constraint such that any violation would be unacceptable, or it can be violated to some extend that should be judged ad hoc by the user? If it is the first case, how to select an appropriate $\lambda$ value would then be crucial, so it would be helpful to mention in the experiments how the authors selected the $\lambda$ such that the constraints were guaranteed not to be violated. If it is the second case, mentioning how much the constraints were violated in individual experiments for a reference.
* Technically, how did the author calculate gradient with respect to the Lagrangian? I suppose this information can be found in the code provided by the author in the supplementary, but I do believe explicitly mentioning this out would be helpful for the potential users of the proposed algorithm.
* As a follow up to the previous question, the authors mentioned that in general the modifications to the environment would be discrete, is this because of the discrete nature of the environment? If it is the case, although the following question would then be out of the scope of this paper, I am wondering how could one obtain the gradient over $w'$ if the environment is continuous?

Minor:
* In the "comparing the accuracy of goal inference" paragraph of page 12, it would be helpful to describe in more detail about how the off-the-shelf Bayesian inference algorithm works, or at least adding a reference, so that the readers would have a better idea about what was done here.

**Strengths And Weaknesses:**

## Strengths
The approach proposed in this paper is conceptually well-explained. The experiments are done throughly to support the authors' claims, especially the real world human dataset collected by the authors.

## Weaknesses
In general, the technical details in this paper should be mentioned more explicitly, and some properties of the proposed approach should also be discussed. See requested changes for more detail. Besides, the authors also mentioned several reasonable limitations of this work in section 6.

---

> ### Author Response · Authors · 2025-07-05
>
> We sincerely thank the reviewer for the thoughtful and detailed feedback. We are pleased that the reviewer appreciates the extensive experimental results, especially the human-subject experiments.  We have uploaded a revised manuscript that incorporates the requested changes, with all modifications highlighted in red for clarity. Below, we address each question and explain how the corresponding points are reflected in the revised manuscript.
>
> **Re: "...as our optimization procedure accounts for the available budget and avoids getting trapped in locally optimal solutions..." — how is this justified given the lack of strong duality?**
> We appreciate the reviewer’s careful attention to this point. We have revised the relevant statement in “Standard Setting” in Section 5.1 to make it more accurate.   When comparing our approach with greedy baselines, we intended to highlight that, unlike greedy methods which make decisions based solely on immediate improvements and are thus more prone to local minima, our approach considers the long-term impact of modifications through gradient-based updates on an objective that accounts for budget constraints. That said, we agree that our method does not guarantee a globally optimal solution either, particularly in the absence of strong duality, and have updated the statement accordingly.
>
> **Re: "...is the budget a hard constraint... or is some violation acceptable? How is λ selected?"**
> In the Lagrangian relaxation, the budget is treated as a soft constraint managed by the Lagrangian multiplier λ. That said, after the optimization process, we can observe the realized budget, i.e., the cost that will be incurred when implementing the changes. In our experiment, we evaluated our method across a wide range of λ values and reported performance based on the realized budget corresponding to each λ. This approach provides a direct and interpretable mapping between λ and the budget, without requiring fine-tuning for each experiment. In practice, if the designer has a specific budget in mind, they can run the optimization with multiple λ values and select the one that yields a realized budget closest to the target. Note that due to the monotonic relationship between λ and the budget (higher λ leads to a smaller budget), binary search–style algorithms can be used to efficiently identify an appropriate λ.
>
> Some of the above discussion on this is currently in “Optimization Procedure” in Section 3.3.  We have expanded the manuscript in “Standard Setting” in Section 5.1 and in Appendix B.2 to provide details on the specific λ parameters that we used in our experiments.
>
> **Re: How is the gradient computed with respect to the Lagrangian?**
> We compute gradients of the loss with respect to w using automatic differentiation via PyTorch’s autograd. We have added the description in Appendix E
>
> **Re: How would the method extend to continuous environments where modifications aren't discrete?**
> The discrete nature of modifications in our current work reflects the structure of the environments used in our experiments (e.g., we can only choose to block the cell or not, and cannot block 30% of the cell). Generally, if we want to extend our framework to a continuous setting, the environment updates could be parameterized by continuous variables, and the budget could be defined over a distance function (e.g., norms) rather than a count of discrete changes.  The Lagrangian term in the loss function would then penalize this continuous deviation, and the gradient would still be computable via the same autograd mechanism. All of our optimization process can be directly applied in the continuous setting: instead of projecting the gradient to the nearest feasible updates, we choose a learning rate and update the environment based on gradient descent. We have included the discussion in “Limitations and Future Work” in Section 6.
>
> **Re: Please elaborate on or cite the Bayesian inference algorithm used on page 12.**
> We appreciate this suggestion. We have provided citations to the Bayesian goal inference method used in our experiments and a footnote describing the approach.

---

### Review · Reviewer_fT8z · 2025-06-14

**Summary Of Contributions:**

This submission introduces a novel data-driven optimization framework for Goal Recognition Design (GRD) that significantly advances the field by addressing key limitations of existing approaches. The core innovation lies in using machine learning to predict "worst-case distinctiveness" (wcd), a metric for how difficult it is to infer an agent's goal. This predictive model is then integrated into a gradient-based optimization framework, enabling efficient modification of decision-making environments to improve goal recognition. The new knowledge presented includes: (1) The machine learning oracle dramatically reduces the computational overhead associated with evaluating wcd, leading to significantly faster runtime compared to traditional exhaustive search methods. (2)  Unlike prior GRD methods that often assume optimal or near-optimal agents, this framework explicitly incorporates diverse agent behavior models, including suboptimal and data-driven human behavior models, making it more realistic and broadly applicable. (3) The approach extends to complex environments like Overcooked-AI and settings with flexible budget constraints, where existing methods are not applicable. (4) The study includes real-world human-subject experiments, demonstrating that the proposed method creates environments that genuinely facilitate efficient goal recognition from human decision-makers, a first for this area of research. This confirms the practical utility and real-world applicability of the framework beyond simulations.

**Audience:**

Yes

**Broader Impact Concerns:**

The work presented in the paper, "Goal Recognition Design for General Behavioral Agents using Machine Learning," primarily focuses on improving the efficiency and applicability of goal recognition design in human-AI collaboration. However, there are a few ethical implications that warrant consideration and could necessitate a broader impact statement: (1) Potential for Manipulation or Unfair Advantage, (2) Privacy Concerns from Behavioral Data, (3) Dependence and Over-reliance: As AI systems become more adept at understanding and influencing human behavior through GRD, there's a potential for humans to become overly reliant on these systems, potentially diminishing their critical thinking or decision-making skills over time.

A broader impact statement would be crucial to acknowledge these potential negative implications and outline steps taken or planned to mitigate them, such as: focusing on transparency, ensuring user control, implementing robust data governance, promoting diverse and representative data collection, and rigorously testing for biases.

**Claims And Evidence:**

Yes

**Requested Changes:**

Critical Adjustments for acceptance:
Discuss Data Scarcity and Scalability to Richer Domains: The paper acknowledges the reliance on "abundant data to train a reliable wcd predictor" and raises concerns about scaling to "richer domains". It is critical to include a brief discussion on potential strategies for addressing data scarcity (e.g., data augmentation, transfer learning, or efficient data collection techniques for real-world scenarios) or how the framework might be adapted when abundant simulation data isn't feasible.

Adjustments to Strengthen the Work:

(1) Address Dynamic and Partially Observable Environments: The paper currently assumes "static, fully observable settings". While mentioned as future work, briefly outlining the conceptual challenges of extending the framework to dynamic and partially observable environments, or suggesting initial research directions, would demonstrate a forward-thinking perspective and strengthen the paper's vision.
(2) Discuss Non-Stationary Agent Behavior: The paper notes that "real human preferences often shift". While acknowledged as a limitation and future work (adaptive or online behavior models), a brief discussion on how the framework could conceptually be adapted to non-stationary human behaviors, perhaps by incorporating adaptive learning components or mechanisms for preference elicitation, would enhance its realism.

**Strengths And Weaknesses:**

Strengths:
Strong Aspects: (1) Addresses Key Limitations of Prior Work: The paper effectively tackles the computational demands and the unrealistic assumption of optimal agent behavior prevalent in existing Goal Recognition Design (GRD) methods. (2) Novel Data-Driven Optimization Framework: The core contribution of leveraging machine learning to predict "worst-case distinctiveness" (wcd) and integrating it into a gradient-based optimization framework is a significant advancement. This approach allows for efficient evaluation and optimization. (3) Improved Runtime Efficiency: The simulations clearly demonstrate superior runtime efficiency compared to traditional exhaustive search and greedy baselines, especially in larger and more complex environments. (4) Human-Subject Validation: The inclusion of human-subject experiments to validate the method's effectiveness with real human decision-makers is a significant contribution and a first for this research area.

Other key strength would be that the paper presents extensive simulations and ablation studies across various settings and agent models, providing strong evidence for the proposed method's effectiveness.

Weaknesses:

(1) Reliance on Abundant Data for Predictive Model: The paper acknowledges that the approach depends on "abundant data to train a reliable wcd predictor". While simulations are used to generate data, the scalability of this data generation to "richer domains" is questioned. Authors could further discuss strategies for data-efficient training or transfer learning in scenarios where abundant real-world data is scarce. (2) Static and Fully Observable Settings Assumption: The current framework assumes "static, fully observable settings". This is a limitation for real-world applications where dynamic and partially observable environments are common. While mentioned as future work, a brief discussion on the challenges and potential initial steps towards addressing these complexities would be beneficial.

---

> ### Author Response · Authors · 2025-07-05
>
> We thank the reviewer for the thoughtful and constructive feedback, and we are pleased that the novelty, practicality, and scope of our contributions were well appreciated. We have uploaded a revised manuscript that incorporates the requested changes, with all modifications highlighted in red for clarity. Below, we address each question and explain how the corresponding points are reflected in the revised manuscript.
>
> **Re: Discuss Data Scarcity and Scalability to Richer Domains**
> We have expanded “Limitations and Future Work” in Section 6 to outline potential strategies for addressing data scarcity and scalability to richer domains.
>
> One promising direction is to leverage transfer learning and fine-tuning, to reduce the data requirements in novel environments. For example, we might train the wcd predictor on readily available simulated data from a set of known environments to capture the structural characteristics of the goal recognition problem. This pre-training phase enables the model to learn generalizable patterns that are not tied to any specific domain. The oracle can then be fine-tuned using limited data from novel environments or human behavior to better capture the nuanced aspects of environments and human decision-making that may not be reflected in simulations.
>
> Additionally, constructing generative models that structurally approximate human behavior in goal recognition tasks might be a possible direction to produce synthetic yet behaviorally plausible training data. These models can serve as an intermediate step to bootstrap the oracle’s learning when real data is limited. Moreover, as suggested by reviewers, leveraging data augmentation or applying active learning techniques to reduce the amount of required data during collection are also promising approaches.
> We also highlight that our framework is modular, and new predictors trained under these constraints can be integrated into the optimization.
>
> **Re: Expand the discussion on dynamic and partially observable environments and non-stationary agent behavior**
> We have expanded  “Limitations and Future Work” in Section 6  to discuss how our framework could be extended to address these limitations. For partial observability, the oracle might be trained on inputs derived from belief state representations, incorporating uncertainty in agent observations directly into the wcd prediction process. For dynamic environments or non-stationary human behavior, the oracle can be periodically fine-tuned using newly accumulated behavioral data, allowing it to adapt to changing patterns in human decision-making over time. This continual adaptation would enhance the model’s robustness in non-stationary settings. Importantly, the optimization component of our framework is modular and remains compatible with such predictors, enabling seamless integration without requiring fundamental changes to the overall method.
>
> **Re: Broader Impact Concerns**
> Thanks for the suggestion. While our work focuses on the positive usage of environment design, we agree that the potential misuse of the techniques should be discussed.  We have added “Broader Impacts” discussion in Section 6 to acknowledge the potential societal impacts.

---

### Review · Reviewer_r8iU · 2025-06-28

**Summary Of Contributions:**

This paper introduces a data-driven framework for goal recognition design (GRD) by learning a predictive model of worst-case distinctiveness (wcd) and using gradient-based optimization to efficiently modify environments, accommodating suboptimal agent behaviors.

**Audience:**

Yes

**Claims And Evidence:**

Yes

**Requested Changes:**

1. How sensitive is the optimization outcome to inaccuracies in the wcd predictor, especially in edge cases?

2. Can the framework generalize to multi-agent settings, or does it fundamentally rely on single-agent assumptions?

3. Are there theoretical guarantees or empirical evidence that the proposed gradient-based optimization avoids poor local minima better than greedy baselines?

4. How would the method perform if applied to continuous action/state spaces, or domains without grid-like structure?

5. What is the role of agent model misspecification (e.g., in human modeling) on the robustness of the learned environment design?

**Strengths And Weaknesses:**

**Strengths**
1. **Clear Motivation and Problem Setup:** The paper addresses two critical limitations of existing GRD methods—computational inefficiency and reliance on optimal agents—and proposes a reasonable ML-based alternative.

2. **Comprehensive Experiments:** The simulations span from simple grid worlds to complex Overcooked-AI environments, showcasing the adaptability and scalability of the proposed method.

3. **Human-Subject Evaluation:** Including human behavior modeling and evaluation adds practical relevance and broadens the contribution.

4. **Runtime Efficiency:** The approach is shown to significantly reduce runtime over traditional exhaustive or greedy methods.

**Weaknesses**
1. **Methodological Simplicity:** The core approach—training a CNN to approximate wcd and using it in gradient-based optimization—is straightforward. There is limited theoretical depth in either the learning or optimization component.

2. **Lack of Theoretical Guarantees:** The paper lacks formal analysis on convergence, generalization bounds of the predictive model, or approximation quality of the learned wcd.

3. **Limited Novelty in Learning/Optimization:** Both the learning (standard supervised regression) and the optimization (basic discrete gradient descent with Lagrangian relaxation) are adaptations of well-known techniques.

4. **Heavy Dependence on Simulated Data:** The method requires a large volume of labeled simulation data for training, which might not generalize to settings where such data is unavailable or unreliable.

5. **Sparse Discussion on Failure Cases:** The limitations of the learned wcd model or optimization getting stuck in local minima are acknowledged but not thoroughly studied.

---

> ### Author Response · Authors · 2025-07-05
>
> We thank the reviewer for the thoughtful and detailed review. We are pleased that the strengths of our work, such as the clear problem formulation, comprehensive experiments across diverse domains, and the inclusion of human-subject evaluation, are recognized. We have uploaded a revised manuscript that incorporates the requested changes, with modifications highlighted in red for clarity. Below, we address your requests and explain how the corresponding points are reflected in the revised manuscript.
>
> **Re: Sensitivity to inaccuracies in the wcd predictor.**
> The optimization outcome of our approach relies on having an accurate wcd predictor. To assess this sensitivity, we conducted an ablation study presented in Section 5.1.1. We compared optimization outcomes using different predictive models (e.g., CNN, Transformer, Linear) trained on datasets of varying sizes for estimating wcd. As shown in Table 2, the performance of the wcd predictor significantly influences the optimization outcome. When coupled with the best predictor, CNN(100K, 0.01), our method outperforms all baselines across settings. However, even when paired with a noticeably less accurate—though still reasonable—predictor, CNN(10K, 0.001), the optimization outcome is comparable to the performance of the baseline that has access to the true wcd. This demonstrates that while our approach benefits from high-quality predictors, it can still achieve competitive performance with reasonably accurate models.
>
> **Re: Generalization to multi-agent settings**
> Our framework can be generalized to multi-agent settings. The key requirement of our approach is the ability to train an oracle that predicts a metric of environment quality, such as wcd, under the specific behavior models of interest. The challenge associated with extending our framework to a multi-agent setting lies in developing agent behavior models to be used in training the oracle, as we need to account for interactions between multiple agents in modeling.  We believe this is an interesting future direction and have added a discussion in “Limitations and Future Work” in Section 6 of the revised manuscript.
>
> **Re: Avoiding poor local minima compared to greedy baselines**
> While we do not have theoretical guarantees, the superior empirical performance of our method compared to greedy baselines suggests that it is less prone to getting stuck in poor local optima. This is primarily because our gradient-based procedure evaluates the global impact of modifications through differentiable updates, rather than committing to myopic and locally beneficial changes in isolation. We have added discussion on this in Section 5.1 subsection “Standard Setting” in the revised manuscript.
>
> **Re: Applicability to continuous spaces and non-grid domains**
> Similar to the multi-agent extension, our framework is generalizable to continuous spaces and non-grid-like domains. The main challenges in this extension are: (1) the increased complexity of agent behavior modeling, as the agent model must now handle continuous state and action spaces, and (2) the potential need for more simulation data to train the wcd predictor in more complex environments. Nonetheless, we believe both challenges are addressable in future work, for example, by using imitation learning for agent modeling in complex environments and transfer learning for data-efficient training. We have included this discussion in “Limitations and Future Work” in Section 6 of the revised manuscript.
>
> **Re: Impact of agent model misspecification**
> Our approach relies on an accurate wcd predictor, which takes the agent behavior model as input. Therefore, if the agent model is significantly misspecified, the resulting wcd predictor may be inaccurate, which in turn can reduce the effectiveness of our environment design, as demonstrated in Section 5.1.1. To address this concern, one potential approach is to adaptively update the predictive model based on observed agent behavior, allowing it to improve over time. We have included notes on this in the revised discussion in “Limitations and Future Work” in Section 6.

---

### Decision · Action_Editor_iPuW · 2025-08-28

**Recommendation:** Accept as is

**Audience:**

Yes

**Audience Explanation:**

The use of human subject experiments alone sets this paper apart from many other works in the field.

**Claims And Evidence:**

Yes

**Claims Explanation:**

The reviewers positively commented on the experimental validation.